



# The Canadian Atmospheric Model version 5 (CanAM5.0.3)

Jason Neil Steven Cole[1], Knut von Salzen[1], Jiangnan Li[1], John Scinocca[1], David Plummer[1], Vivek Arora[1], Norman McFarlane[1], Michael Lazare[1], Murray MacKay[2], Diana Verseghy[3], and Barbara Winter[1]

[1]Canadian Centre for Climate Modelling and Analysis, Environment and Climate Change Canada, Victoria, British Columbia, Canada
[2]Environmental Numerical Weather Prediction Research Section, Environment and Climate Change Canada, Toronto, Ontario, Canada
[3]Climate Processes Section, Environment and Climate Change Canada, Toronto, Ontario, Canada

*Correspondence to:* Jason Cole (jason.cole@ec.gc.ca)

**Abstract.** The Canadian Atmospheric Model version 5 (CanAM5) is the component of Canadian Earth System Model version 5 (CanESM5) which models atmospheric processes and coupling of the atmosphere with land and lake models. Described in this paper are the main features of CanAM5, with a focus on changes relative to the last major scientific version of the model (CanAM4). These changes are mostly related to improvements in radiative transfer, clouds and aerosol parameterizations, as

well as a major upgrade of the land surface and land carbon cycle models and addition of a small lake model. In addition to changes to parameterizations and models, changes in the adjustable parameters between CanAM4 and CanAM5 are documented. Finally, the mean climatology simulated by CanAM5 for present day are evaluated against observations and compared with that simulated by CanAM4. Although many of the aspects of the simulated climate are similar between CanAM4 and CanAM5, there is a reduction of precipitation and temperature biases over the Amazonian basin, global cloud fraction biases,

and solar and thermal cloud radiative effects, all of which are improvements relative to observations.

## 1 Introduction

The fifth version of the Canadian Atmospheric Model (CanAM5) is a major component model in the Canadian Earth System Model version 5 (CanESM5) (Swart et al., 2019), modelling atmospheric processes and coupling of the atmosphere with land and lake models. Both CanAM5 and CanESM5 are models developed by the Canadian Centre for Climate Modelling

and Analysis (CCCma) to simulate climate to improve understanding and make predictions and projections of future climate. CanAM5 is the result of several years of development on its last major scientific version, CanAM4 (von Salzen et al., 2013), which was the atmospheric component of CanESM2 (Arora et al., 2011) used for CMIP5 (Taylor et al., 2011). Between CanAM4 and CanAM5, there were numerous changes to radiative transfer, cloud and aerosol parameterizations, plus a major upgrade of the land surface model and addition of a model of unresolved subgrid-scale lakes.

CanESM5 and CanAM5 were the basis for CCCma's contribution to the sixth Coupled Model Intercomparison Project (CMIP6) (Eyring et al., 2016), which included a number of experiments to better understand and characterize cloud feedbacks and radiative forcings. The mean state and response of CanESM5 to external forcing in fully coupled simulations is docu-




mented in Swart et al. (2019). In this paper, the focus is on the ability of CanAM5 to simulate the historical climate for AMIP simulations in which sea surface temperature and sea ice concentration are prescribed from observations. Since there are several changes in CanAM5 relative to CanAM4, most of the evaluation with observations is performed with both CanAM5 and CanAM4 to highlight changes in the simulated climate.

In Sections 2 and 3, changes in atmospheric and surface processes between CanAM4 and CanAM5 are presented, respectively. In Section 4 the process used to tune CanAM5 is discussed and the values used for adjustable parameters presented. In Section 5, the experiments used to evaluate CanAM5 are presented and the details of their setup are described. In Section 6, an analysis of climatological features of CanAM5 are presented and prominent biases discussed. Finally, in Section 7 we conclude with a brief summary and discussion of the main results of this study.

**2   Atmospheric processes**

This section summarizes atmospheric parameterizations in CanAM5. As most parameterizations are described and documented in detail for CanAM4 (von Salzen et al., 2013) we focus on the major changes between CanAM5 and CanAM4.

The horizontal resolution of CanAM5 is identical to CanAM4 and is defined by triangular truncation at total wavenumber of 63 (ie T63). The model employs a double spectral transform allowing the physical tendencies to be evaluated on a reduced

"linear" T63 Gaussian grid of dimensions $128 \times 64$, which corresponds to $\sim 2.8°$. The number of vertical levels in CanAM5 has increased from 35 to 49. The 49 levels are used with layer thicknesses that increase monotonically from approximately 100 m at the surface to 2 km at $\sim 1$ hPa which is the upper bound of the vertical domain. The the additional 14 layers in CanAM5 have been added to the upper tropopause and lower stratosphere to match those those employed by the Canadian Middle Atmosphere Model (Scinocca et al., 2008).

**2.1   Radiation**

Radiative transfer in CanAM5 includes new specification of optical properties for the surface, cloud, and aerosol, in addition to computation of radiative fluxes accounting for sub-gridscale surface varability.

The parameterized absorption by gases uses a correlated $k$-distribution model that is mostly unchanged from CanAM4, using the same wavelength intervals and quadrature points (von Salzen et al., 2013). A significant modification is the addition of a

solar water vapour continuum (Mlawer et al., 1997), which resulted in improved simulation of absorption at solar wavelengths (Pincus et al., 2015). The single scattering properties of the ice clouds in CanAM4 were parameterized for thermal and solar wavelengths under the assumption that all the ice particles are hexagonal prisms (von Salzen et al., 2013). In CanAM5 the optical properties of ice particles use the parameterization of Yang et al. (2012), assuming a mixture of ice habits that is based on spaceborne observations and assuming a moderately rough surface, which is found to improve retrievals (Baum et al., 2011).

The single scattering properties of pure liquid clouds remain the same but can be perturbed to account for internally mix black carbon (Li et al., 2013) allowing simulation of the semi-indirect effect.





Aerosol optical properties in CanAM5 use updated single scattering properties as well as an improved methods to mix aerosol optics. The single scattering properties for organic carbon use the refractive index from HITRAN 2012 (Rothman et al., 2013) and the properties of black carbon from Flanner et al. (2012). Instead of externally mixing aerosols, it is assumed that sulfate and the hydrophilic components of black carbon and organic carbon are internally mixed (Wu et al., 2018). The

refractive index of the internally mixed aerosol is computed based on the fraction, effective radius, and effective variance of each component aerosol, as well as relative humidity, which is used to compute hydrophilic growth.

The ocean optical properties are changed in CanAM5. In CanAM4, the whitecap albedo was wavelength invariant with a value of 0.3. In CanAM5, this was replaced with mean values (0.216, 0.134, 0.044, 0.005) for each wavelength interval used in the solar radiative transfer calculations based on Frouin et al. (2001). The parameterization of ocean albedo is similar to

that in CanAM4 but the contents of the lookup table has been updated and now includes a dependence on the solar zenith angle and partitioning of the downwelling solar radiation into direct and diffuse components (Jin et al., 2011). This partitioning is estimated using the vertically integrated aerosol/cloud optical depth. In CanAM4 the ocean albedo was computed using as input optical depths and solar zenith angles from the last radiative transfer timestep which is 1 hour earlier. To improve the consistency of the ocean albedo calculation and the radiative transfer calculations, especially near sunrise and sunset, in

CanAM5 the ocean albedo is calculated using cloud and aerosol information from the previous dynamical timestep, 15 minutes earlier, and the solar zenith angle from the current timestep.

Over land, the dry bare soil albedo in CanAM4 was set to a global constant combined with a parameterization to account for the effect of surface wetting (Verseghy, 2012). The use of a globally constant bare soil albedo resulted in regional biases for clear-sky albedo at the top of atmosphere, especially over the Sahara and Australian interior. The constant albedo was replaced

with a regionally varying soil colormap and associated albedos (Lawrence and Chase, 2007). These new location dependent bare soil albedo map greatly reduced biases in clear-sky albedo relative to observations. In addition to the bare soil, the albedo and emissivity of snow and sea ice was also updated. The albedo of snow on land and sea ice in CanAM5 is computed using a lookup table accounting for snowpack properties including snow water equivalent, snow grain size, and black carbon simulated by the land surface model (Namazi et al., 2015). The emissivity of snow present on land or sea ice uses a single, wavelength

invariant emissivity, which was reduced from 1.0 to 0.97 (Chen et al., 2014). Similiarly, the emissivity of sea ice was reduced from 1.0 to 0.97 to be consistent with the CanESM5 sea ice model (Fichefet and Maqueda, 1997).

Within a CanAM5 gridbox, there can be multiple surface types including land, lake, ocean, and sea ice (Sect. 3.3). When coupling the atmosphere with ocean, sea ice and land models, it is necessary to have surface radiative fluxes that are consistent with each surface type. While it is possible to partition the grid-mean radiative fluxes at the surface using the tiled albedo,

emissivity and temperature, in CanAM5 radiative flux profiles are instead computed for each surface type and then averaged to a grid mean. It is assumed that the same atmosphere is present over each tile. Although this approach causes a small increase (<5%) in the total time for global radiative transfer calculations in a CanAM5, it maintains consistency between the surface and the atmosphere. The modest increase in computational time is because multiple surface tiles are only present in a portion of the CanAM5 gridboxes, e.g., there is only one surface type over sea ice free ocean.





## 2.2 Aerosols and chemistry

The types of natural and anthropogenic aerosols in CanAM5 include sulfate, black and organic carbon, sea salt, and mineral dust, similar to CanAM4 (von Salzen et al., 2013). Parameterizations for aerosol emissions and transport, gas-phase and aqueous-phase chemistry, as well as dry and wet deposition account for interactions with simulated meteorology. Natural aerosol species are represented in the model using prognostic emission fluxes. In particular, a particle-size dependent emission scheme is used to account for aeolian erosion in arid and semi-arid regions (Peng et al., 2012). Sea salt concentrations in two size modes are parameterized as a function of the wind speed near the surface of the ocean (Ma et al., 2008). Dimethyl sulfide emissions are predicted using specified climatological concentrations in the surface ocean (Tesdal et al., 2016a, b). Sulfur oxidation in the gas and aqueous phases is simulated using specified climatological oxidant concentrations from CMAM20 (McLandress et al., 2013).

The chemistry parameterizations in CanAM5 are unchanged in CanAM5 with the exception of stratospheric water vapour which can be produced by methane oxidation using a parameterization based on that described in ECMWF (2003).

## 2.3 Clouds

The parameterization of clouds and cloud microphysics in CanAM5 is mostly the same as in CanAM4 (von Salzen et al., 2013). Like CanAM4 and other global climate models, CanAM5 continues to employs bulk cloud microphysical parameterizations which depend on mean water content and other moments of the droplet size distribution.

For liquid clouds in CanAM4 the autoconversion parameterization of Khairoutdinov and Kogan (2000) is used and is replaced in CanAM5 with the the autoconversion parameterization from Wood (2005). The parameterization of Wood (2005) is a modification of the parameterization by Liu and Daum (2004) to simulate the collision and coalescence of cloud droplets. For convenience we reproduce here the main equations for Khairoutdinov and Kogan (2000) and Wood (2005) since adjustment of parameters in the equations is discussed in Sec. 4.

Autoconversion in CanAM4 is parameterized as (Khairoutdinov and Kogan, 2000)

$$\frac{\partial q_r}{\partial t} = A \overline{q_L^{2.47}} N_d^{-1.79} \rho^{-1.47}, \tag{1}$$

where $A$ is a constant, $q_r$ is the rain water content (in kg m$^{-3}$), $q_L$ is the liquid cloud water content (in kg m$^{-3}$), $N_d$ the droplet number concentration (in m$^{-3}$), and $\rho$ is air density (in kg m$^{-2}$). In CanAM5, the autconversion is parameterized as (Wood, 2005)

$$\frac{\partial q_r}{\partial t} = E \overline{q_L^3} N_d^{-1} H(R_6 - R_{6C}), \tag{2}$$

where $q_r$ is the rain water content (in kg m$^{-3}$), $q_L$ is the liquid cloud water content (in kg m$^{-3}$), $N_d$ the droplet number concentration (in m$^{-3}$) and $H()$ is the Heaviside function. Additionally, $E = 1.3 \times 10^9 \beta_6^6$, $R_{6C} = 7.5/(\overline{q_L}^{1/6} R_6^{1/2})$, $R_6 = \beta_6 r_v$ and $\beta_6 = [(r_v + 3)/r_v]$, where $r_v$ is the mean volume radius (in $\mu$m). In order to account for the impacts of sub-gridscale variability in cloud liquid water content, the statistical cloud scheme in CanAM5 is used to determine the mean value of $q_L^3$, indicated by the bar in Eq. 2.



Along with the new autoconversion parameterization, the second aerosol indirect effect is active in CanAM5 (it was not in CanAM4). Given the uncertainty of applying the parameterizations at high altitudes, clouds processes are limited to pressures greater than 10 hPa.

## 3 Surface processes

There were three substantial changes to the treatment of surfaces processes. A major upgrade of the land surface including tighter integration of the land carbon cycle model, a small lakes model and tiling to accommodate fractional land in a gridbox.

### 3.1 CLASS-CTEM

The land component of CanAM5 is represented by the Canadian Land Surface Scheme (CLASS) and the Canadian Terrestrial Ecosystem Model (CTEM) which model physical and biogeochemical processes, respectively. In the CanAM5 version 3.6 of
CLASS is used which models the energy and moisture fluxes at the air/land-surface interface (Verseghy, 2012).

Compared to its predecessor used in CanAM4 (CLASS 2.7 (Verseghy, 1991; Verseghy et al., 1993)) there are several major structural improvements in version 3.6 of CLASS. These include optional implementation of a user-specified number of soil layers rather than the previous hard-coded three layers, and the capability of supporting a mosaic of vegetation, soil, water or ice tiles within grid cells. The capability of modelling fully organic soils has been introduced, with hydraulic properties
assigned on the basis of the work of Letts et al. (2000). The thermal conductivities of the organic and mineral soil layers are determined following Côté and Konrad (2005) and Zhang et al. (2008). The wet and dry albedos of the mineral soil are assigned based on a global soil reflectivity index described in Lawrence and Chase (2007) and Oleson et al. (2010). Organic soil albedos are assigned following Comer et al. (2000). The bare soil surface evaporation efficiency parameter is calculated using a relation presented by Lee and Pielke (1992). Empirical corrections are applied to the saturated hydraulic conductivity
of each soil layer to take into account the viscosity of water at the layer temperature (Dingman, S., L., 2002) and the presence of ice Zhao and Gray (1997). The field capacity of the lowest permeable soil layer and the baseflow at the bottom of the layer are obtained using relations derived from Soulis et al. (2011).

A new option is provided to model snowpack albedo and transmissivity in four wavelength intervals instead of two intervals (Jason Cole, personal communication) The thermal conductivity of snow is obtained from the snow density using a relationship
derived by Sturm et al. (1997). The fresh snow density is calculated as an empirical function of the air temperature, using relations developed by Hedstrom and Pomeroy (1998) and Pomeroy, John Willard and Gray, D. M. (1995). The maximum snowpack density is calculated as a function of snow depth following Tabler et al. (1990). The amount of snowfall intercepted by vegetation, and the unloading rate of intercepted snow, are calculated following Hedstrom and Pomeroy (1998). The canopy interception capacity for snow is determined from the plant area index and the fresh snow density as described in Bartlett et al.
(2006) and Schmidt and Gluns (1991). Albedos of snow-covered vegetation canopies are set following Bartlett and Verseghy (2015). The sensible and latent heat fluxes between the vegetation, the underlying ground and the overlying atmosphere are evaluated based on the analysis of Garratt, J. R. (1992), which incorporates an explicit treatment of the canopy air space.



Although CLASS 3.6 can represent vegetation as a mosaic, the composite approach of representing different vegetation types in a grid cell is employed in CanAM5. This implies that area-weighted grid-mean structural vegetation attributes of different vegetation types are used in energy and water balance calculations. The number of soil and bedrock layers remains three and the same as in CanAM4 with the first and second soil layers 0.1 and 0.25 m thick. The maximum thickness of the permeable

soil for the third layer is 3.75 m but it varies geographically depending on the permeable soil depth specified following Zobler,L (1986).

CTEM models vegetation as a dynamic component of the climate system and provides structural attributes of vegetation to CLASS for use in its physics calculations (Arora and Boer, 2005). These include leaf area index, vegetation height, rooting depth and distribution, and canopy mass. The biogeochemical component CTEM has not changed much from CanAM4 except

the diagnostic calculation of wetland extent and methane emissions (Arora et al., 2018) none of which affect the physical land surface processes.

## 3.2   Canadian Small Lake Model

CanAM5 includes a parameterization for subgrid-scale lakes to improve surface fluxes of heat and moisture over land masses. The scheme is based on the Canadian Small Lake Model, CSLM (MacKay, 2012; MacKay et al., 2017). This scheme computes

a nonlinear surface energy balance in a thin skin layer and then solves the heat equation based on thermal conduction and shortwave radiation extinction following Beer's law for both visible and near infrared bands. A diurnal surface mixed layer is simulated based on the bulk turbulent kinetic energy approach, e.g., Niiler and Kraus (1977), developed by Rayner (1980), Imberger (1985), and Spigel et al. (1986) for lakes. A seasonal thermocline arises naturally as a result of the daily excursions of the surface mixed layer. The equation of state follows Farmer and Carmack (1981), except that the effects of pressure and

salinity are neglected.

The model allows for the formation of both black, i.e. congelation, and white ice. Black ice forms when the energy balance in a layer is sufficiently negative to cool it below 0°C. White ice forms when the weight of the overlying snowpack is sufficient to crack the ice and allow lake water to flood a layer of snow, which is then assumed to freeze immediately and completely. Latent heat from the freezing of the pore water is first used to warm the snow crystals in the slush layer to 0°C, with the

remainder going into the overlying snowpack. Both white and congelation ice are assumed to be free of air bubbles and to have the same transmissivity.

Fractional ice cover, following Leppäranta and Wang (2008), and fractional snow on ice are permitted, thus allowing for the simultaneous presence of open water, bare ice, and snow covered ice. Fractional ice cover is especially important for larger lakes subject to sufficient wind stress, which can mechanically break ice to produce pressure ridges and open water leads. The

presence of some open water will alter turbulent and radiative flux exchange with the atmosphere, as well as light availability at depth due to differences in roughness, albedo, and light extinction between water and ice. Snow itself is represented as in the Canadian Land Surface Scheme CLASS (Sec. 3.1), with the snowpack simulated as a layer thermally distinct from the underlying ice.



The properties and interaction of all lakes within a CanAM5 grid cell are modelled by one representative sub-grid lake using CSLM. The properties of the representative lake in each CanAM5 grid cell are derived from the Global Lake Database version 2 (GLDv2) (Kourzeneva et al., 2012; Choulga et al., 2014), which is provided at $1/120° \times 1/120°$ resolution. The grid fraction covered by the representative lakes is derived from the aggregate area of lakes in GLDv2 that falls within each CanAM5 grid

cell. This defines the unresolved lake tile (Sec. 3.3). Lake dynamics are governed by three external geophysical parameters that must be specified: the visible light transparency, mean depth (or volume), and mean fetch. For all representative lakes, a constant transparency of 0.5 m$^{-1}$ is assumed and the mean depth and mean fetch in each CanAM5 grid cell are derived in an aggregate manner from GLDv2.

## 3.3 Tiling

To more easily facilitate conservative coupling, all previous versions of coupled atmosphere-ocean models developed at CC-Cma employed coincident grids with an identical binary land mask, e.g., CanESM2 employed CanAM4's land mask for its CMIP5 contribution. In these earlier versions, enhanced ocean resolution was achieved by prescribing multiple ocean grid cells below each CanAM atmospheric grid cell. For CanESM5, independent arbitrarily oriented grids are assumed for both the atmosphere and the ocean (Swart et al. (2019)) This required the implementation of a fractional land mask in the atmospheric

model and tiling of its underlying surface. In general, each CanAM5 grid cell can contain tiles representing land, ocean, sea ice and unresolved lakes. The tiling approach used is a generalization of that discussed in Sec. 3.1 for the tiling of vegetation types over the land portion of atmospheric grid cells. For example, within each model grid cell, independent energy and water fluxes are derived over each underlying surface type given its unique properties, e.g. temperature, albedo etc.. On the atmospheric side, these fluxes undergo a weighted aggregation based on the tile fraction to produce a single flux seen by the atmosphere. In

fully coupled mode, if ocean and/or sea ice sits below some portion of an atmospheric grid cell, the flux of each representing each surface type is passed to the coupler, CanCPL, and is remapped and transferred to the underlying grid of the ocean and/or sea ice model.

Currently, surface tiling in CanAM5 has been implemented for parameterization of radiative transfer, surface processes, and vertical diffusion. Aside from the radiation, the fluxes over each tile are derived from the prognostic variables in the lowest

model level, e.g. temperature, specific humidity, wind, etc.. For simplicity, the blending height at which the fluxes from each tile are aggregated is also taken to occur in the lowest atmospheric model level. For radiative transfer calculations, profiles of radiative fluxes are computed for each of the tiles and aggregated into a grid mean while fluxes are maintained for each tile as described in Sec. 2.1.

## 3.4 Snow on sea ice

For snow on sea ice in CanAM5, the parameterization of snow cover fraction was updated and a parameterization of wet snow grain growth added, both for more consistency with the treatment of snow on land.



In CanAM4, different parameterizations of snow cover were used on land and on sea ice, the snow cover on land being (Verseghy, 2012),

$$X_{snow} = \begin{cases} Z_{snow}/Z_{snow,lim} & \text{if } Z_{snow} \leq Z_{snow,lim} \\ 1.0 & \text{if } Z_{snow} > Z_{snow,lim} \end{cases} \tag{3}$$

and snow cover over sea ice, being

$$X_{snow} = \begin{cases} \sqrt{SWE/SWE_{lim}} & \text{if } SWE \leq SWE_{lim} \\ 1.0 & \text{if } SWE > SWE_{lim} \end{cases} \tag{4}$$

where $X_{snow}$ is the fractional area of the land or sea ice covered with snow, $Z_{snow}$ is the depth of the snow (in m), $SWE$ is the snow water equivalent (in kg m$^{-2}$) with $Z_{snow,lim}$ and $SWE_{lim}$ being adjustable limits for each. In CanAM5, Eq. 3 is used to determine snow cover over land and sea ice. Note that in Eq. 3, $Z_{snow}$ is initially computed using $Z_{snow} = SWE/\rho_{snow}$, where $\rho_{snow}$ is the snow density in kg m$^{-3}$. If $Z_{snow} \leq Z_{snow,lim}$ then $Z_{snow} = Z_{snow,lim}$ and the $SWE$ adjusted accordingly (Verseghy, 2012).

The computation of sea ice albedo includes a contribution from snowpack on sea ice when it is present. To calculate the albedo of snow, it is necessary to simulate the relevant physical properties of the snow, including the snow grain size. The approach used to parameterize these properties in CanAM5 is described in Namazi et al. (2015). Described here is the addition of a parameterization to CanAM5 so that wet growth of snow grains is included for snow on sea ice, where previously only dry snow grain growth was considered.

To calculate the wet growth of snow grains, the same expression is used as over land (Eq. 3 of Namazi et al. (2015)) which requires knowledge of the liquid water fraction in the snowpack, which was not available. In CanAM5, the snowpack liquid water fraction is parameterized using (Anderson, 1976),

$$F_{liq} = \begin{cases} F_{liq,min} & \text{if } \rho_{snow} \geq \rho_{snow,thres} \\ F_{liq,min} + (F_{liq,max} - F_{liq,min}) * \frac{\rho_{snow,thres} - \rho_{snow}}{\rho_{snow,thres}} & \text{if } \rho_{snow} < \rho_{snow,thres} \end{cases} \tag{5}$$

where $F_{liq}$ is the fraction of liquid in the snow pack, $F_{liq,max}$ and $F_{liq,min}$ the maximum and minimum allowed values of $F_{liq}$, $\rho_{snow}$ the density of snow (in kg m$^{-3}$) and $\rho_{snow,thres}$ the snow density threshold at which $F_{liq,max}$ occurs.

## 4 Setting adjustable parameters

After finalizing the new and updated physical parameterizations for CanAM5 were finalized, they were held fixed, or frozen, and only a subset of parameters were manually adjusted within a range of physically plausible values to target a stable and reasonable climate in the coupled atmosphere-ocean model CanESM5. This is the last exercise performed to finalize a model version and is often referred to as "tuning". The subset of parameters and values in CanAM5 is provided in Table 1. They include parameters adjusted specifically for CanAM5 and parameters adjusted when tuning intermediate versions of CanAM



between versions 4 and 5, e.g., CanAM4.1. In this section, we discuss the process used to arrive at the values, which differed from that used in CanAM4/CanESM2.

The tuning of CanESM2 was carried out mainly by adjusting the parameters of each of its components separately, including CanAM4, with a goal of minimal additional adjustments when fully coupled. For example, the parameters for CanAM4 were

mostly tuned using transient prescribed SST and sea ice simulations of near-present day, consistent with simulations used regularly for CanAM development. Applying the same approach to tuning of CanESM5 resulted in a coupled preindustrial (1850) control simulation with a climate that was too cold with excessive sea ice relative to observations. Therefore, CanAM5 was tuned in the context of fully coupled CanESM5 simulations, with a particular focus on obtaining preindustrial control conditions with global mean temperatures and sea ice within acceptable ranges. The combination of parameters that achieved

this target was then scrutinized to verify that other aspects of the climate remained acceptable. Analyses to investigate the effect of parameter sets on the climate included CanAM5 simulations using prescribed SSTs and sea ice (AMIP). For the most part, the mean climate simulated in AMIP mode was close to coupled CanESM5 simulations with the noticable exception of the net radiative flux at the top of atmosphere (TOA). Adjustments required to ensure an acceptable preindustrial climate resulted in values of TOA ($\sim 2.5\,\mathrm{W\,m^{-2}}$) in historical AMIP runs, which are larger than observations. However, the value of

the TOA net radiative flux in CanESM5 historical coupled simulations during present day are very close to those observed (e.g. $\sim 1\,\mathrm{W\,m^{-2}}$). Details of the TOA radiative fluxes are discussed in Sec. 2.1 but for the purposes of tuning this particular bias in AMIP simulations was retained to get a reasonable preindustrial control climate.

Table 1 lists parameters that changed between CanAM4 and CanAM5, with ones in bold specifically adjusted for CanAM5 and others having been changed when tuning intermediate CanAM versions between CanAM4 and CanAM5. This table does

not include parameters that were adjusted in the ocean or in the sea ice model that only affected coupled simulations. The rightmost column of Table 1 provides sources and, where possible, references that explain the setting in CanAM. This final set of CanAM5 parameters allows us to simulate a climate that is on balance reasonable relative to observations in both coupled and AMIP mode.

The parameters related to cloud microphysics have notable effects on radiative energy budgets and coupled climate, in-

cluding emergent CanESM5 properties such as climate sensitivity. Of particular importance are the two parameters scaling the efficiency of cloud droplet autoconversion and accretion to precipitation. In CanAM5, the accretion rate factor was the main parameter adjusted instead of autoconversion, which is opposite to the approach used when tuning CanAM4. Analysis of satellite observations by Lebsock et al. (2013), indicate global climate models may severely underestimate mean accretion rates when subgrid cloud-precipitation covariability is omitted. Furthermore, Gettelman et al. (2015), Sant et al. (2015), and

Michibata et al. (2019) showed that diagnostic parameterizations of rain processes, such as those employed in CanAM5, produce considerably lower accretion rates than prognostic and more comprehensive parameterizations. Consequently, the usual assumptions of an instantaneous and horizontally uniform precipitation flux in the cloudy portions of the grid cells in CanAM5 likely cause unrealistically low accretion rates. In an attempt to compensate for this, the original parameterization of accretion of Khairoutdinov and Kogan (2000) is made more efficient through the considerable increase (factor of 15) of the tunable

parameter. Autoconversion rates, on the other hand, are not scaled.



**Table 1.** Adjustable parameters in CanAM and their settings in CanAM5. The values in **bold** were specifically tuned in CanAM5, while the others were used to tune intermediate versions of CanAM. The leftmost column indicates references that discuss the adjustable parameter or include further references about the parameter.

| Scheme | Parameter | Physical description | CanAM5 | unit | Comment/Reference |
|---|---|---|---|---|---|
| Cloud micro-physics | facacc | factor scaling mass accretion rate of cloud water to precipitation due to the collection of cloud droplets by raindrops | **15** | $s^{-1}$ | Wood (2005) |
| | facaut | factor scaling efficiency coefficient in mass autoconversion rate of cloud water to precipitation due to the collision-coalescence processes of cloud droplets | **0.1204** | – | Khairoutdinov and Kogan (2000) |
| | uicefac | factor to scale ice crystal fall speed due to the influence of gravity | **6000** | $s^{-1}$ | von Salzen et al. (2013) |
| Moist con-vection | alf | proportionality parameter relating vertically integrated convective kinetic energy with the cloud base mass flux | $5.0 \times 10^8$ | $m^4\,kg^{-1}$ | Scinocca et al. (2008) |
| | ccu | weight large-scale and pressure gradient force contributions to moist convection horizontal velocity (updrafts) | 0.0 | – | von Salzen et al. (2013) |
| | ccd | weight large-scale and pressure gradient force contributions to moist convection horizontal velocity (downdrafts) | 0.0 | – | von Salzen et al. (2013) |
| Gravity wave | fcrit | critical inverse Froude number | 0.22 | – | Phillips (1984) |
| | gphil | mountain sharpness number | 1.0 | – | McFarlane (1987) |
| Vertical diffu-sion | rkhmn | minimum background vertical diffusivity for temperature | 0.1 | $m^2\,s^{-1}$ | von Salzen et al. (2013) |
| | rkqmn | minimum background vertical diffusivity for moisture | 0.1 | $m^2\,s^{-1}$ | von Salzen et al. (2013) |
| Surface pro-cesses | drn | scaling factor for soil drainage at the bottom of the soil levels | 0.1 | – | Verseghy (2012) |
| | cuscale | scaling factor of the wind stress threshold for dust emissions | 1.6 | – | Peng et al. (2012) |
| | reff0_sea | background specific surface area of snow grains (on sea ice) | **30** | $m^2\,kg$ | Personal communication (Joshua King) |
| | reff0_land | background specific surface area of snow grains (on land) | **60** | $m^2\,kg$ | Personal communication (Joshua King) |
| | albp | depth of melt ponds on sea ice | **20** | cm | Ebert and Curry (1993) |



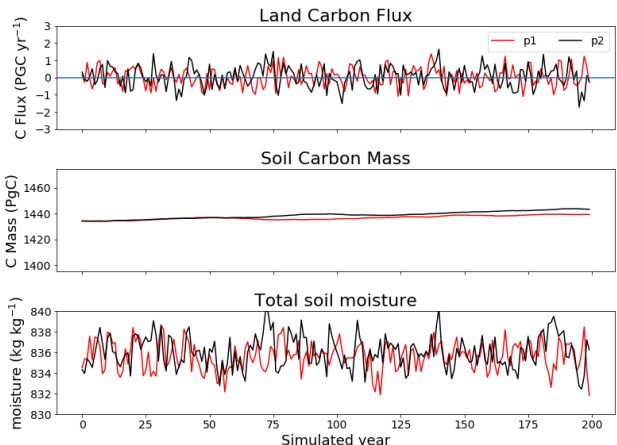

**Figure 1.** Land carbon fluxes, soil carbon mass, and total soil moisture for years 300-500 of the CanAM5 1870 control simulation. The red and black lines show, respectively, the results for CanAM5 using two different physics configurations of CanESM5, p1 and p2, the details of which are described in Swart et al. (2019).

## 5 Control and CMIP6 simulations

Unlike CanAM4, CanAM5 has an interactive land carbon cycle (Sec. 3.1) which necessitates starting CanAM5 transient simulations from a state with a land carbon cycle that is reasonably close to equilibrium. To achieve this, sufficiently long simulations with a stable climate are required. This is done using an approach similar to the spinup of the CanESM5 preindustrial simu-

5 lation (Swart et al. (2019)). A long control simulation of CanAM5 is performed using a repeating annual cycle of forcing and prescribed sea surface temperature (SST) and sea ice. For this CanAM5 control, we use forcing for year 1870 (the first year in the historical SSTs and sea ice dataset), while the annual cycle of SST and sea ice is the mean over 1870-1879.

With this configuration, the CanAM5 control simulation was initialized from a coupled CanESM5 1850 control simulation and run for ∼300 years until land properties, including carbon, approached a new quasi-equilibrium. The simulation was then

10 extended by an additional 200 years. Figure 1 shows the land carbon flux and the total soil moisture during the 200 years. Most atmospheric variables reached a quasi-equilibrium within a few years and are therefore not shown. For CanESM5, transient coupled simulations were started from the control coupled simulation every 50 years (Swart et al., 2019). A similar approach was used for CanAM5 with transient simulations starting from the CanAM5 control simulation every 10 years beginning at year 400. These are used to generate a ten member ensemble using transient forcings, SSTs, and sea ice for the period 1870 to

15 2014.

Several CMIP6 experiments were performed using CanAM5 and prescribed SSTs and sea ice. The ten member ensemble of 1870-2014 transient simulations were contributed to the Global Monsoon Model Intercomparison Project (Zhou et al., 2016), while the period 1950 to 2014 from each simulation makes up the CanESM5 contribution to the DECK AMIP experiment (Eyring et al., 2016). DECK AMIP simulations are the basis for several CFMIP experiments used to characterize and under-



## 6 Evaluation of CanAM5 climatology

The properties of coupled atmosphere-ocean experiments using CanESM5 are shown in (Swart et al., 2019). Documented in this section is the climatology of CanAM5 from the CMIP6 AMIP simulation, evaluated against observations and highlighting differences compared to CanAM4 from a CMIP5 AMIP simulation. Details of observations used for evaluation are summarized in Table A1 and model variables are summarized in Table A2. For all figures, the first ensemble member is used for each AMIP simulation, r1i1p1 for CanAM4 and r1i1p2f1 for CanAM5. For the most part, the results using prescribed SSTs and sea ice are similar to coupled CanESM5 and CanESM2 simulations. For ease of comparison with CanESM5 coupled simulations, the figures in this section have been reproduced in the Supplemental for the first member of the ensemble (r1i1p2f1) of the CanESM5 historical simulation (Swart et al., 2019).

### 6.1 Clouds and precipitation

The near-global (equatorward of 60°) cloud fraction as a function of cloud optical thickness and cloud top pressure is shown in Fig. 2. For the purposes of consistent model output from CanAM5, CanAM4, and observations, model output from the International Satellite Cloud Climatology Project (ISCCP) simulator (Bodas-Salcedo et al., 2011) are compared with ISCCP observations, both ISCCP-D (Rossow and Schiffer, 1999) and ISCCP-H (Knapp et al., 2021). The two version of ISCCP observations are used to illustrate the uncertainty in the cloud properties, an uncertainty that only increases once other cloud observations are considered (Stubenrauch et al., 2013). For example, Pincus et al. (2015) showed that there are large differences between ISCCP and MODIS (larger than between the two versions of ISCCP shown here). Therefore, it is important that such differences be considered in the evaluation of models, especially for optically thin clouds.

With these caveats in mind, the histograms of biases indicate that CanAM5 generally overestimates the amount of cloud with moderate optical thickness of high- and low-altitude clouds and undersimulated clouds at mid-level altitudes. Looking at clouds as a function of visible optical thickness, there is a shift to more optically thin ($\tau < 23$) in CanAM5. The structure of these biases relative to ISCCP is consistent with previous studies, for example, (Klein et al., 2013).

The zonal mean structure for cloud amount is presented (Fig. 3) which illustrates that the near-global mean biases are due to regional biases and informs us of the source of biases and improvements in the cloud radiative effect (CRE) (Fig. 7). As seen in the near-global means, the differences between ISCCP-D and ISCCP-H are generally smaller than biases between CanAM and ISCCP-H and the change in biases between CanAM4 and CanAM5. Although there remain biases in the total cloud amount, there is a systematic reduction of biases in CanAM5 of approximately 3%. Parsing the biases in CanAM5 by the altitude of cloud top pressure, the increase in the CanAM5 total cloud amount is caused by both low (cloud top pressure > 680 hPa) and non-low (cloud top pressure < 680 hPa) cloud amount. There is a notable increase in Southern hemisphere



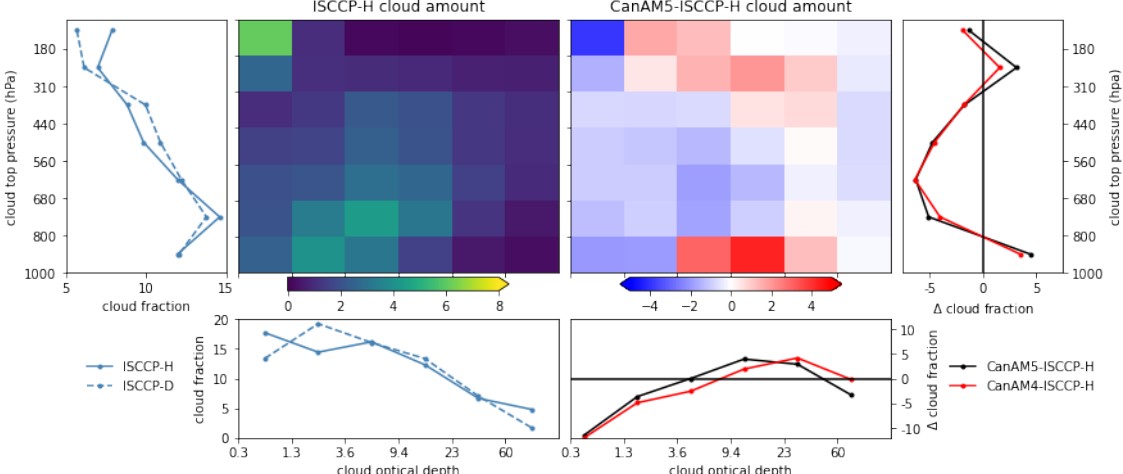

**Figure 2.** Mean histograms of the cloud fraction equatorward of $60°$ as a function of the cloud top pressure and cloud visible optical thickness from ISCCP-H and the biases in CanAM5 (mid panels in the upper row). To the side of each histogram is the mean cloud fraction, or cloud fraction bias, as a function of cloud top pressure, while below each histogram is shown the cloud fraction, or cloud fraction bias, as a function of cloud optical thickness. Means are averages for 1987-2008.

low cloud in CanAM5 and a systematic increase in the amount of "thin" (cloud visible $\tau$ between 0.3 and 23), consistent with the near-global mean. The increase in cloud amount is consistent with the change in (CREs) (Fig. 7). While the shift to more optically thin cloud would reduce their reflectively it does so in a non-linear manner instead of the linear response of CRE to changes in cloud fraction.

The CMIP6 protocol requested the additional diagnostic output consistent with retrievals based on lidar observations from CALIPSO (Chepfer et al., 2010) and MODIS imager measurements (Pincus et al., 2012). These are used to evaluate the vertical structure of the clouds in CanAM5 and the ability of CanAM5 to simulate the cloud phase, Fig. 4. Biases in the cross-section of cloud amount for CanAM5 relative to GCM-Oriented CALIPSO Cloud Product (GOCCP), upper row Fig. 4, is consistent with biases between CanAM4 and GOCCP (von Salzen et al., 2013). There is too much cloud simulated at high altitudes and too little cloud at low altitudes, except for the cloud nearest the surface, which is similar to observations or has too much cloud present.

The middle and lower rows of Fig. 4 use diagnostics of GOCCP cloud phase profiles and MODIS cloud top phase. In the tropics, CanAM5 underestimates the fraction of cloud that is ice in the middle troposphere, however, it occurs in a range of altitudes where CanAM5 is already simulating too few clouds. The more notable bias is that CanAM5 simulates too much ice cloud poleward of $\sim 50°$, seen in the GOCCP and MODIS diagnostics. These biases in the high-latitude cloud phase can have important consequences on the radiation budget and cloud feedbacks in these regions (Storelvmo et al., 2015).

Precipitation biases are an important feature of any climate model. Although the structure of the biases in CanAM5 is similar to that in CanAM4, there are improvements in some key regions. The most noticeable improvement is the increased precipi-




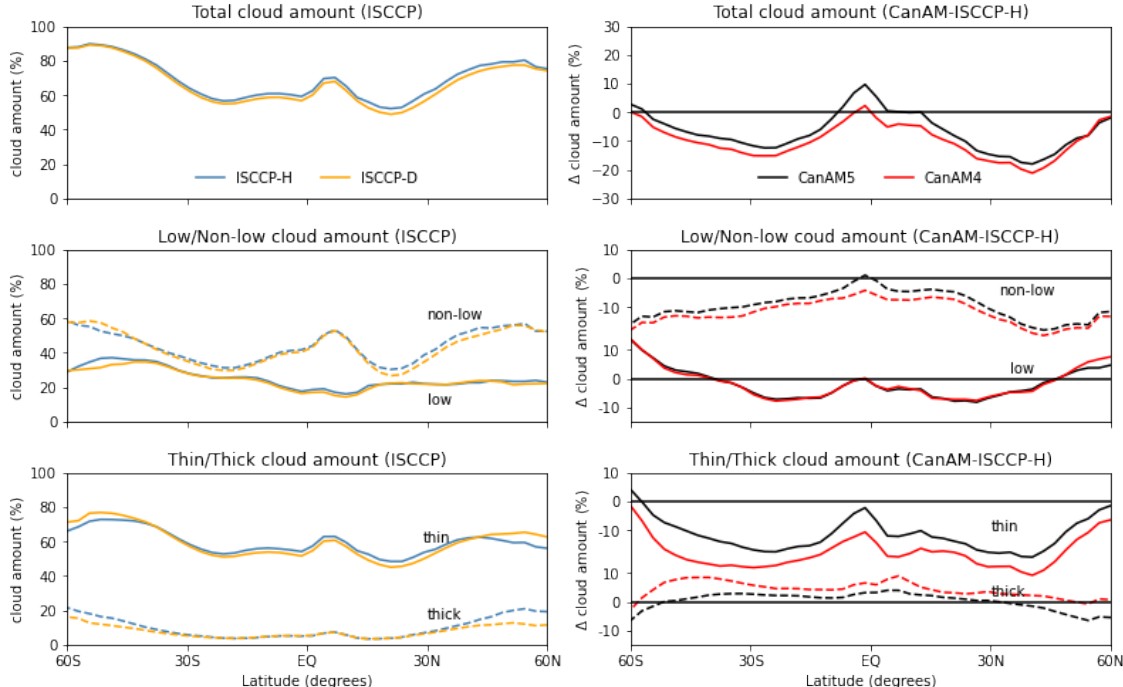

**Figure 3.** Zonal mean cloud fraction for the total cloud amount (upper row), cloud amount for low (cloud top pressure > 680 hPa) and non-low (cloud top pressure < 680 hPa) in the middle row, and the cloud amount for thin (cloud visible $\tau$ between 0.3 and 23) and thick (cloud visible $\tau$ > 23) in the bottom row. All CanAM and ISCCP cloud amounts only consider clouds with $\tau$ > 0.3. Observations for ISCCP-H and ISCCP-D are shown in the left column, and biases for CanAM5 and CanAM4 relative to ISCCP-H in the right column.

tation rate over the Amazon in CanAM5 for most seasons, although dry biases remain, Fig. 5. This change in precipitation is consistent with a reduction in too warm temperatures over the Amazon in CanAM5, Fig. 11, which may be due to more moist conditions suppressing the near-surface temperature.

## 6.2 Radiation

Radiative fluxes through the top of atmosphere (TOA) and bottom of the atmosphere are evaluated using CERES observations. Figure 6 summarizes the global mean climatology for the solar and thermal flux components of the radiative energy budget for CanAM5 and CanAM4. Although a relatively short common period is used from the models and CERES observations (2003-2009), the results are very similar to those using longer periods from CERES (2003-2020) and CanAM (1979-2009).

We focus first on fluxes at TOA, which can be most directly compared with space-based observations from CERES. The global mean thermal fluxes are effectively identical in CanAM4 and CanAM5. The change in the downward solar flux is due to the use of updated solar forcing (Matthes et al., 2017) that has a reduced total solar irradiance, which is more consistent with



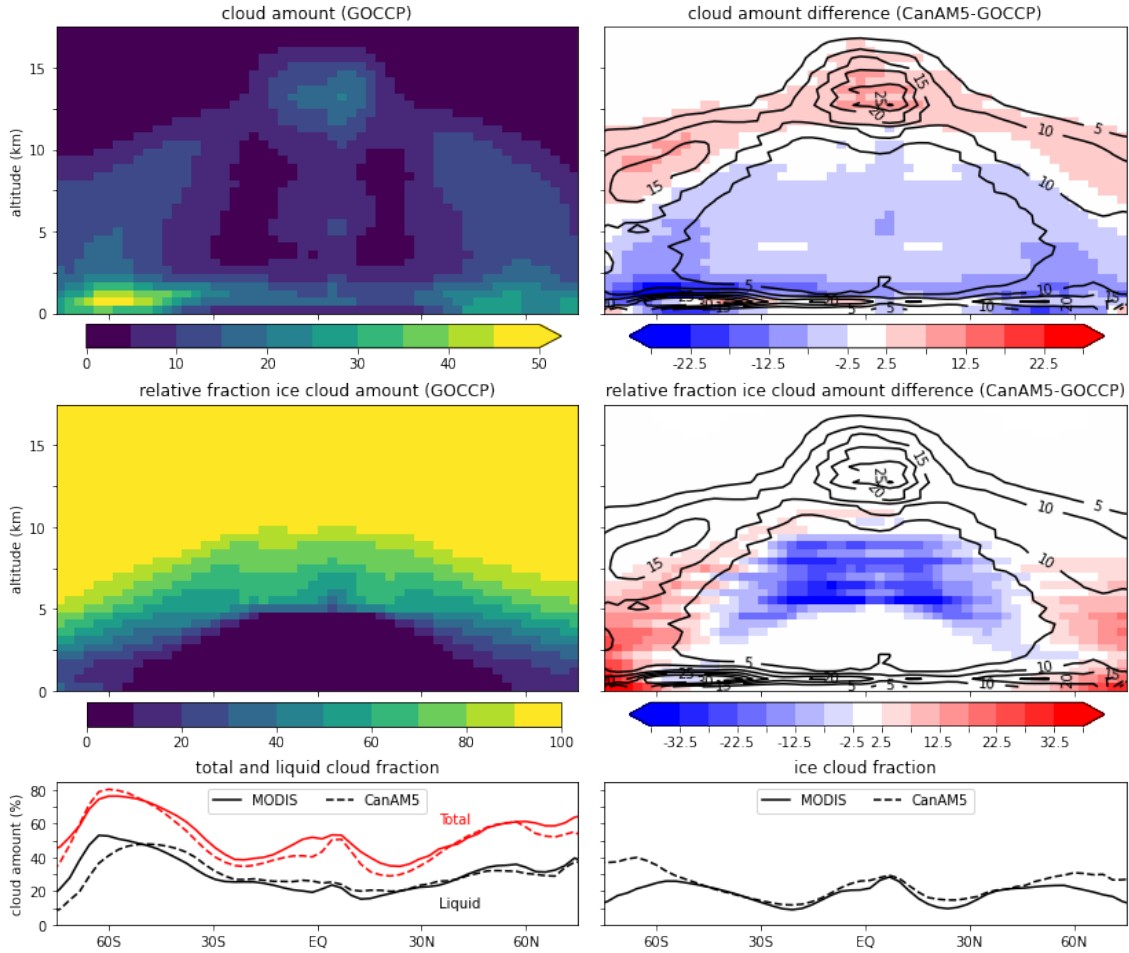

**Figure 4.** Zonal cloud fraction and cloud phase from CanAM5 compared with GOCCP and MODIS obervations. Means are averages for 2007-2009.

observations and CERES. The upward solar flux is reduced by 3.4 W m$^{-2}$. There is a small reduction in the clear-sky upward solar flux at TOA, $\sim 0.2$ W m$^{-2}$, so the remainder of this reduction is due to clouds (Fig. 7) .

Evaluated separately, the solar and thermal radiative fluxes are within the range of values from the CMIP6 simulations (Wild, 2020). When all fluxes are combined to compute the net flux imbalance at the TOA, CanAM5 has a value that is larger than CERES and CanAM4 by 2.2 W m$^{-2}$. In CanAM4 there is a compensation between the upward thermal (too small) and upward solar (too large) resulting in a net imbalance that is in line with observations while in CanAM5 both the upward thermal and solar are smaller than observations. We note that at least one other model that participated in CMIP6 found a similar difference between AMIP and coupled simulations (Hourdin et al., 2021).

The TOA flux imbalance is larger than that for coupled CanESM5 simulations, $\sim 1.1$ W m$^{-2}$, averaged over 2003-2009. This is mainly due to solar fluxes which are larger, 99.3 W m$^{-2}$, than that when using observed SSTs, 97.7 W m$^{-2}$, since upward



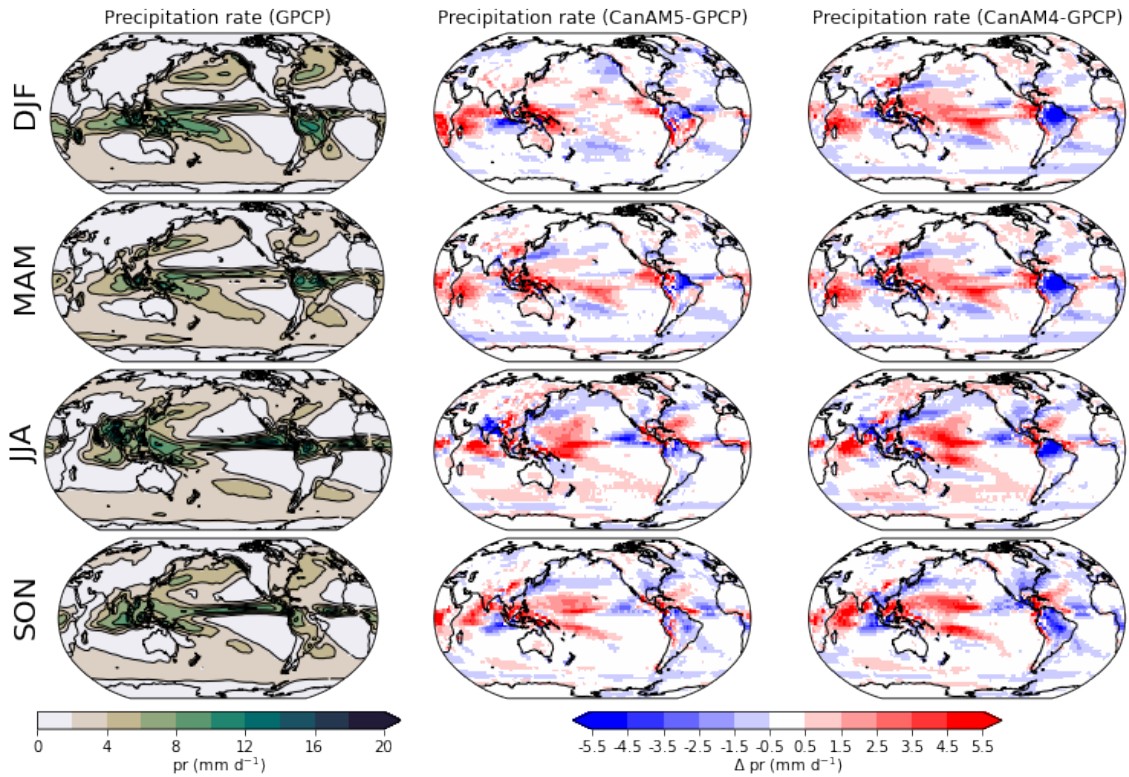

**Figure 5.** Seasonal mean precipitation rate from GPCP (left column), the bias of CanAM5 relative to GPCP (middle column) and the bias of CanAM4 relative to GPCP (right column). Means are computed from years 1980-2009.

thermal fluxes at TOA are similar, 239.8 W m$^{-2}$ versus 239.5 W m$^{-2}$. An in-depth analysis of why this occurs is beyond the scope of this paper. Preliminary analysis using CanAM5 with combinations of sea ice and SST specification, from observations and coupled CanESM5 simulations, suggests that differences in SSTs (Swart et al., 2019) are the main factor which may be due to local and nonlocal responses affecting the TOA radiative fluxes.

5    While the TOA radiative fluxes were regularly evaluated during development of CanAM, radiative fluxes at the surface and within the atmosphere were not. For both CanAM5 and CanAM4 the biases at the surface relative to CERES are consistent: too little downward and upward longwave flux, too large downward and upward shortwave fluxes. All together this results in too little absorption of radiation at the surface by 3 to 5 W m$^{-2}$ and fairly consistent overestimation of net absorption in the atmosphere by 5 W m$^{-2}$ due mainly to too much absorption of longwave radiation.

10    Clouds strongly modulate radiative fluxes so we next examine the simulated cloud radiative effects (CREs), defined as $CRE = F_{clearsky} - F_{allsky}$, where $F_{clearsky}$ is the radiative flux in the absence of clouds and $F_{allsky}$ is the radiative flux with clouds present. The annual mean cloud radiative effects are generally positive for longwave and negative for shortwave the TOA and surface, while the longwave strongly controls the zonal atmospheric CRE (Fig. 7). The global mean CREs simulated



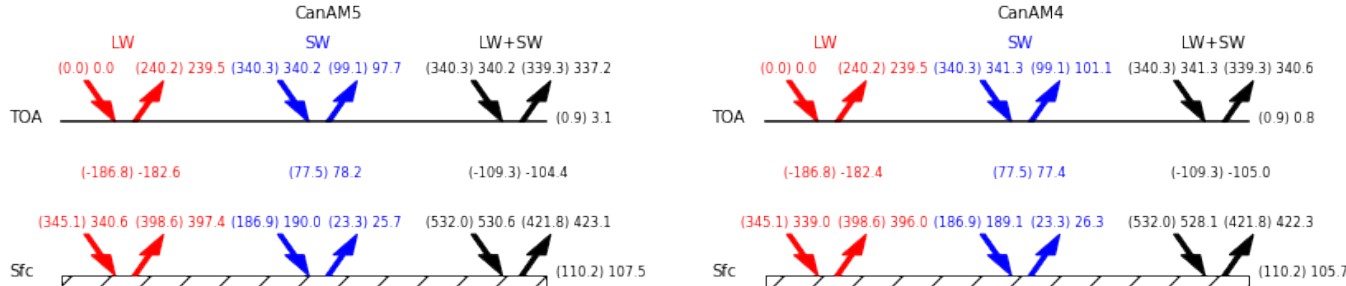

**Figure 6.** Global and time mean radiative fluxes at the top of atmosphere (TOA) and surface, as well as the net flux divergence for the atmosphere, from AMIP simulations by CanAM5 and CanAM4 compared with CERES EBAF. For each pair of numbers, the left is CERES and the right is CanAM5 or CanAM4. The means are averages over 2003-2009.

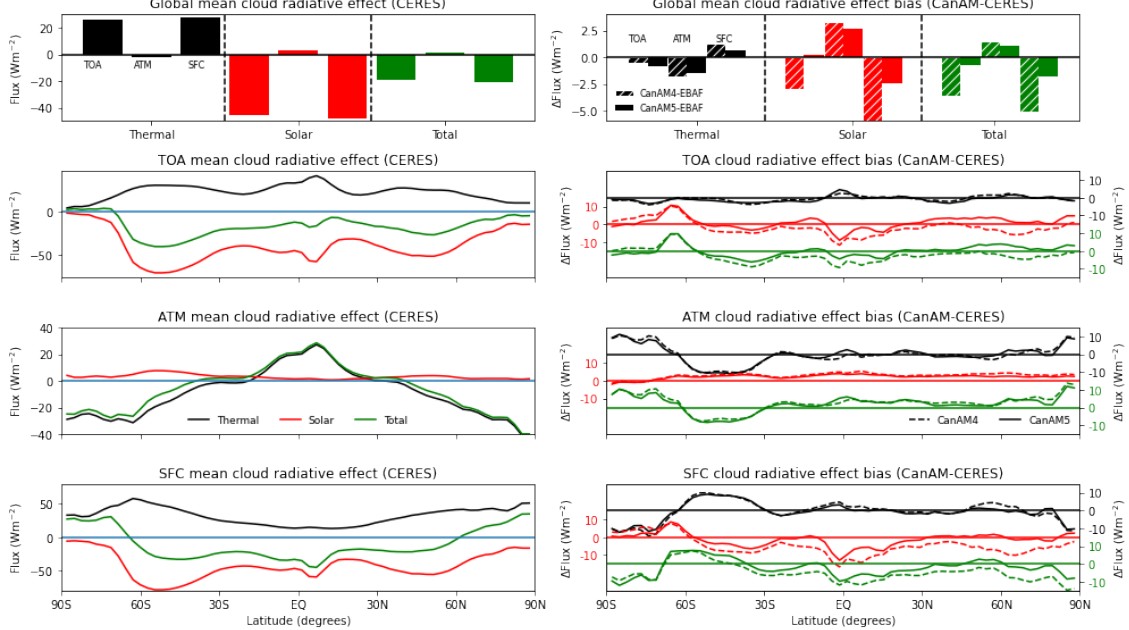

**Figure 7.** Annual global and zonal mean cloud radiative effects at the top of atmosphere, surface, and atmosphere, from CERES EBAF observation (left column) and from CanAM5 and CanAM4 AMIP simulations (right column). The means are averages over 2003-2009.

by CanAM5 are less biased relative to CERES than CanAM4, especially for shortwave CRE, while the longwave CRE at TOA is slightly more biased than CanAM4, Fig. 7. Zonal mean CREs show that the improvements seen in the CanAM5 global means are due to reduced biases at most latitudes, Fig. 7. These improved CREs suggest improved simulation of cloud properties in CanAM5.





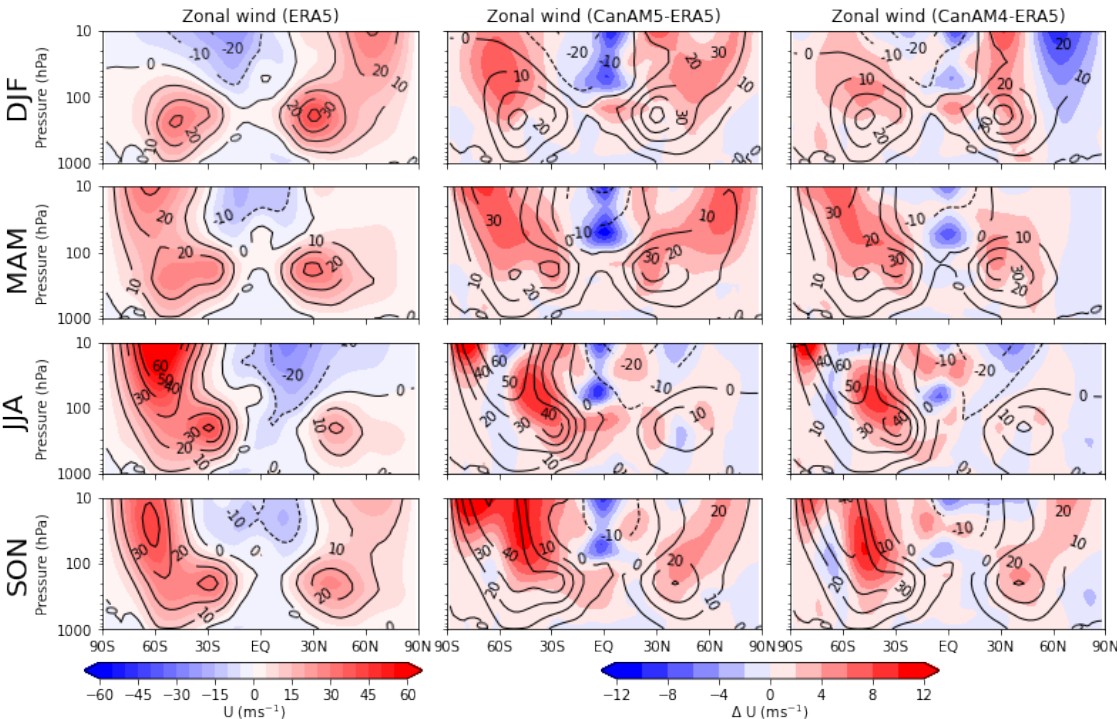

**Figure 8.** Seasonal mean latitude-pressure plots of zonal wind from ERA5 (left column), the bias of CanAM5 relative to ERA5 (middle column), and the bias of CanAM4 relative to ERA5 (right column). For all plots, contours are the mean. For the ERA5 plot, shading is the mean, in other plots the shading is the bias relative to ERA5. All plots use data from years 1980-2009.

## 6.3  Circulation

In this subsection we document the climatological properties of the winds, temperature, and surface pressure in CanAM5 for an AMIP experiment. Seasonal climatologies of latitude-height zonal-mean zonal wind fields and anomalies are presented in Fig. 8. While overall biases are similar, relative to CanAM4, CanAM5 displays anomalously positive rather than negative
5  wind biases in the mid- to high-latitude Northern Hemisphere DJF lower stratosphere. This apparently results in a weaker planetary wave forcing of the northern hemisphere stratosphere in CanAM5. This positive zonal wind bias is associated with changes to adjustable parameter values in the orographic gravity-wave drag parameterization between the two model versions, Sec. 4. The near-surface zonal wind climatology is consistent with the coupled CanESM5 simulations (Swart et al., 2019), with biases relative to ERA5 generally smaller in CanAM5, with the most significant reductions coming in midlatitudes in
10  both hemispheres, Fig. A1.

In Fig. 9, seasonal climatologies of latitude-height zonal-mean temperature from ERA5, CanAM5, and CanAM4 are presented. In general, CanAM5 and CanAM4 have similar patterns of temperature bias in all seasons, including a warm tropical





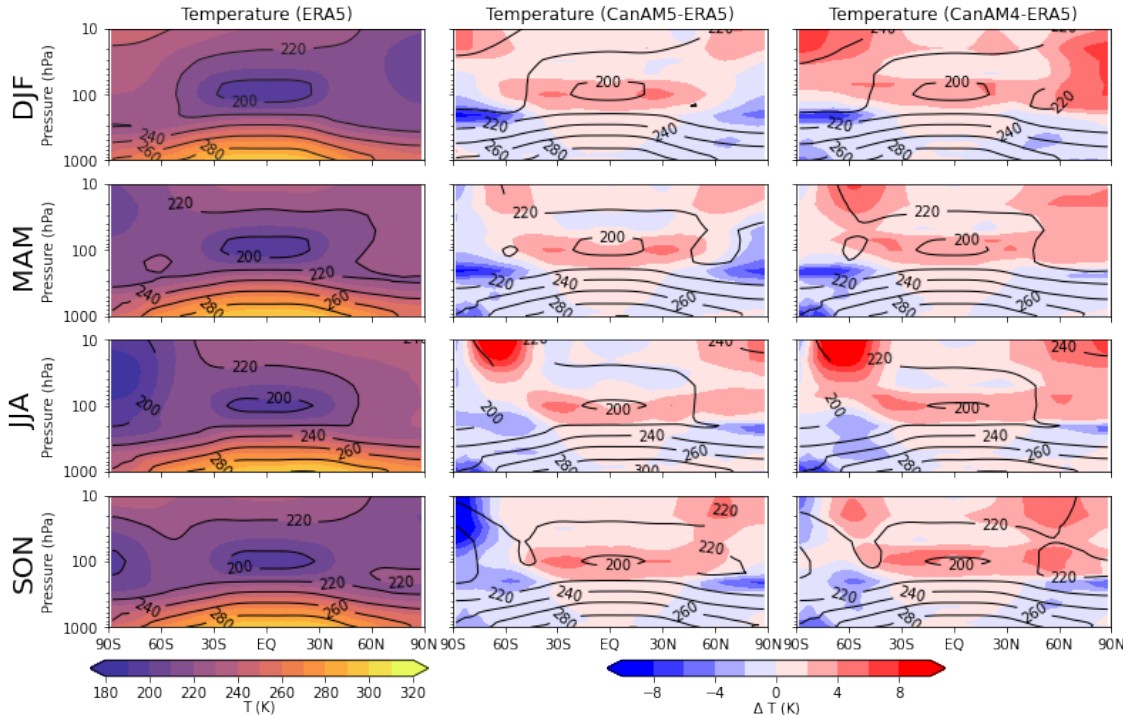

**Figure 9.** Seasonal mean latitude-pressure plots of temperature from ERA5 (left column), the bias of CanAM5 relative to ERA5 (middle column) and the bias of CanAM4 relative to ERA5 (right column). For all plots, contours are the mean. For the ERA5 plot, shading is the mean, in other plots the shading is the bias relative to ERA5. All plots use data from years 1980-2009.

tropopause and cool extratropical tropopause. The pattern and magnitude of the temperature biases are similar to those in coupled configurations (Swart et al., 2019).

The seasonal-mean sea-level pressure is presented in Fig. 10. Relative to CanAM4, CanAM5 displays larger DJF biases in the Aleutian low and North Pacific high, but lower bias in the whole of the Atlantic Ocean in all seasons.

5    Seasonal mean biases in near-surface temperature for CanESM5 and CanESM2 are presented in Fig. 11. Persistent cold biases are found over the Tibetean plateau and North Africa in both models. The Tibetian plateau bias is negatively correlated with snow cover bias (too much snow cover and too cold temperatures), a feature found in other CMIP6 models (Lalande et al., 2021). The source of too large snow cover is complex and is present to differing degrees in CanAM4 and CanAM5. That said, it does seem to be a robust feature of CanAM, given that the land model was significantly changed between CanAM4 and

10  CanAM5, including the parameterization of snow albedo, Sec. 3.1. In North Africa, cold biases are thought to be due to the change from a globally constant albedo for bare soil to a more realistic distribution based on local soil conditions (Lawrence and Chase, 2007).

Warm biases are apparent over the Brazil basin, although somewhat reduced in CanAM5, consistent with biases related to too little precipitation, Fig. 5. Over central North America in JJA, the warm bias persists in CanAM5 and is more extensive



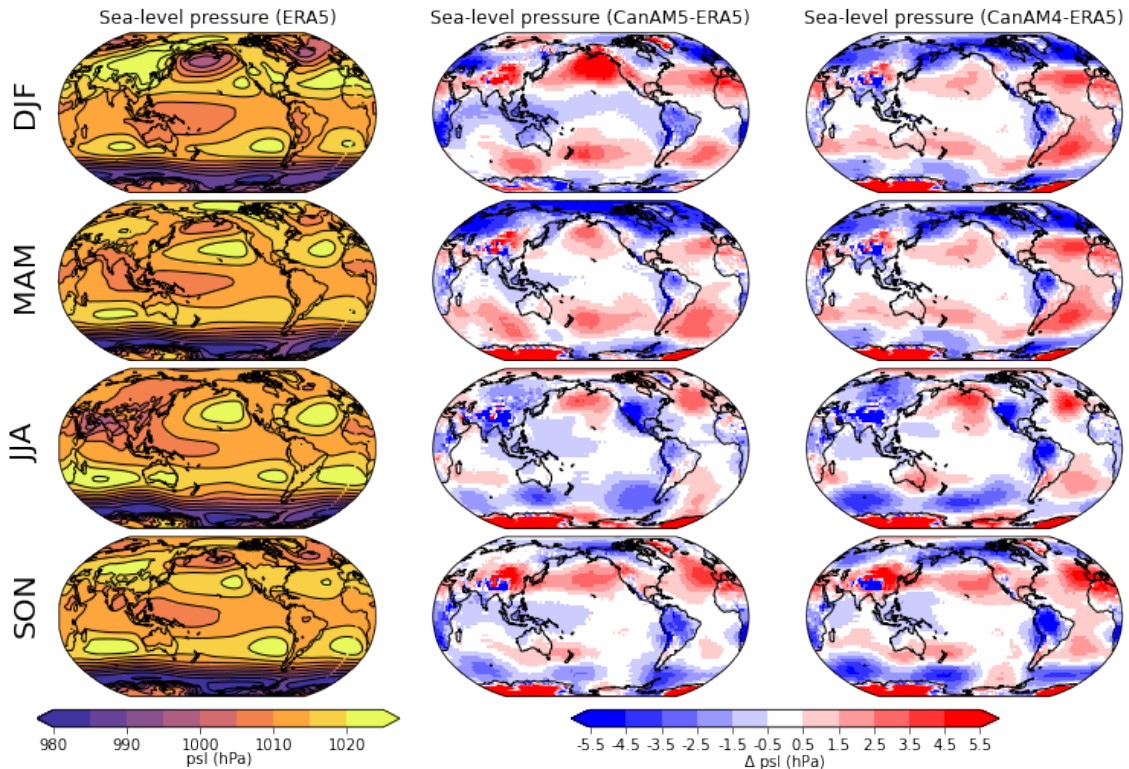

**Figure 10.** Seasonal mean sea-level pressure from ERA5 (left column), the bias of CanAM5 relative to ERA5 (middle column) and the bias of CanAM4 relative to ERA5 (right column). All plots use data from years 1980-2009.

than in CanAM4. This is a common warm bias among CMIP5 models during JJA (Cheruy et al., 2014) for which the cause is thought to be a complex interplay between land-atmosphere coupling, radiation, and clouds that rapidly develop in climate models (Morcrette et al., 2018).

# 7   Conclusions

5   CanAM5 is the latest atmospheric model from the Canadian Centre for Climate Modelling and Analysis. In this study, we have presented the main developmental differences between CanAM5 and its predecessor and CanAM4. In particular, these differences are primarily related to radiation, clouds, and aerosols, a major update of the land surface model, and the addition of a parameterization of fresh-water lakes. Generally, mean climatologies from CanAM5 for near-present conditions, and using observed SSTs and sea ice, are similar to those from CanAM4, with some notable improvements, reduced precipitation and

10   temperature biases over the Amazonian basin, reduced cloud fraction biases, and reduction in solar and thermal CREs. Some biases persist from CanAM4 to CanAM5, e.g. cold biases over the Tibetian Plateau, and new biases are present in CanAM5



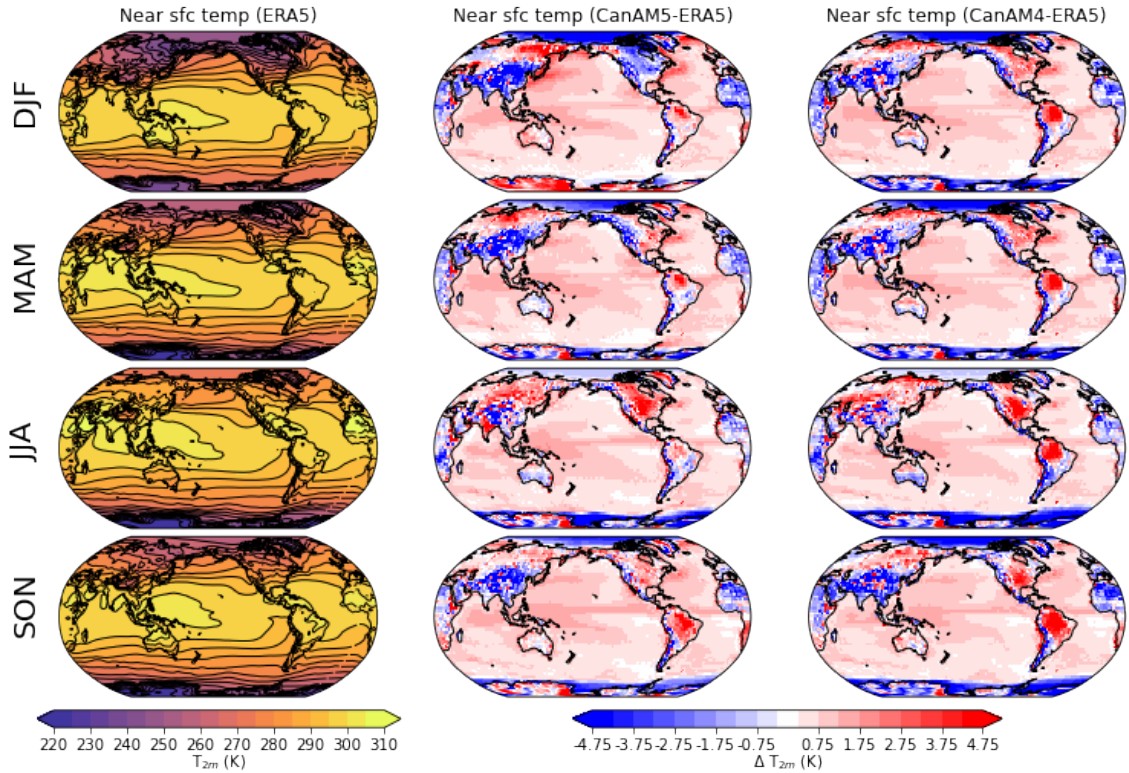

**Figure 11.** Seasonal mean near surface temperature from ERA5 (left column), the bias of CanAM5 relative to ERA5 (middle column) and the bias of CanAM4 relative to ERA5 (right column). All plots use data from years 1980-2009.

when using prescribed SSTs and sea ice, e.g., a bias in net downward flux at TOA. As noted, the bias in the net downward flux at TOA was done purposely to have a reasonable 1850 control coupled simulation using CanESM5.

Why it was necessary to tune the net downward flux at TOA higher than observations when using observed SSTs and sea ice remains a question for further research. Additional simulations with CanAM5, using combinations of observed SSTs and

5    sea ice with SSTs and sea ice from coupled CanESM5 simulations, suggest this is due to the pattern of SSTs in CanESM5. This was not the case for CanESM2 and CanAM4 which could be largely tuned for coupling using observed SSTs and sea ice. Further analysis, including the use of Green's functions (Zhou et al., 2017) to link regional differences in SSTs to global mean fluxes at TOA, should help inform future tuning of CanESM and CanAM. Another question considered, is why CanESM5 has a significantly larger climate sensitivity than CanESM2, and nearly all CMIP6 models (Zelinka et al., 2020). At present,

10   this thought to be change is mostly due to changes in cloud feedbacks (Virgin et al., 2021). This suggests that improved mean climatologies of clouds and radiation in CanAM5 and CanESM5 do not necessarily result in improved cloud feedbacks (Zelinka et al., 2022) and climate sensitivity. Better understanding of both these questions will provide guidance for ongoing development of CanAM.



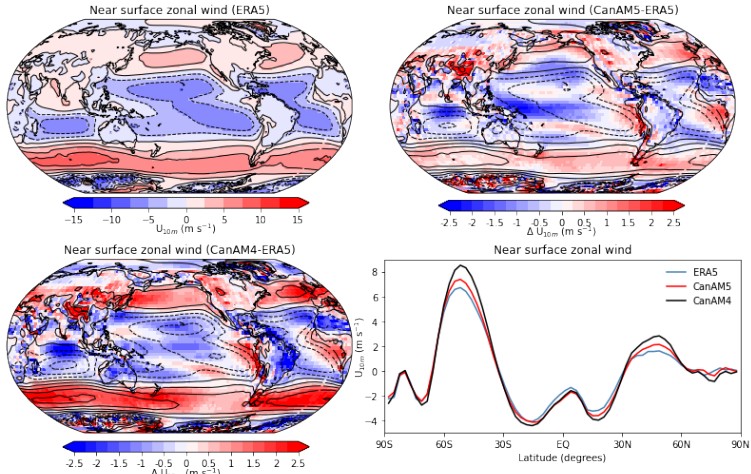

**Figure A1.** Annual mean near surface zonal wind, nominally 10 m above the surface from ERA5 (upper left) and CanAM5 and CanAM4 biases. For all plots, contours are the mean, while for the ERA5 plot shading are also the mean but in the other plots the shading is the bias relative to ERA5. All plots use data from years 1980-2009.

*Code and data availability.*  The full CanESM5 source code is publicly available at https://gitlab.com/cccma/canesm and includes CanAM5 as a submodule. The version of the code which can be used to produce all simulations submitted to CMIP6 and described in this manuscript is tagged as v5.0.3, and has the associated DOI: 10.5281/zenodo.3251113. The scripts used to produce all the figures are available at https://doi.org/10.5281/zenodo.7579680. All CanESM5/CanAM5 and CanESM2/CanAM4 simulations conducted for CMIP6 and CMIP5, respectively, including those described in this manuscript, are publicly available via the Earth System Grid Federation (ESGF). All observational data used are publicly available and are listed in Table A1.



**Table A1.** Observational data used for model evaluation.

| Data | Description | Version | Reference |
|---|---|---|---|
| TOA radiative fluxes | CERES EBAF-TOA | 4.1 | Loeb et al. (2018) |
| Surface radiative fluxes | CERES EBAF | 4.1 | Kato et al. (2018) |
| Lidar based cloud amount | GOCCP (3D_CloudFraction) | 3.1.2 | Chepfer et al. (2010) |
| Lidar based cloud phase | GOCCP (3D_CloudFraction_phase) | 3.1.2 | Cesana and Chepfer (2013) |
| Cloud amount histogram | ISCCP H (HGG) | v01r00 | Knapp et al. (2021), Rossow et al. |
| Cloud amount histogram | ISCCP D | 2 | Rossow and Schiffer (1999) |
| Cloud top phase | MODIS | 6 | Pincus et al. (2012) 10.5067/MODIS/MCD06COSP_D3_MODIS.061 |
| Atmospheric and surface data | ERA | 5 | Hersbach et al. (2020) Hersbach et al. (2019a), Hersbach et al. (2019b) |



**Table A2.** CMIP6 and CMIP5 data used for model evaluation.

| Figure number | CMIP6/CMIP5 variable | Description |
|---|---|---|
| Figure 2 | clisccp | Histogram of cloud amount by cloud top pressure and cloud visible optical thickness |
| Figure 3 | clisccp | Histogram of cloud amount by cloud top pressure and cloud visible optical thickness, consistent with ISCCP. |
| Figure 4 | clcalipso, clcalipsoliq, clcalipsoice, clwmodis, climodis, cltmodis | Cloud profile consistent with CALIPSO and cloud fraction from MODIS. |
| Figure 5 | pr | Precipitation rate |
| Figure 6 | rsdt, rsut, rlut, rsds, rsus, rlds, rlus | Radiative fluxes at top of atmosphere and surface. |
| Figure 7 | rsdt, rsut, rsutcs, rlut, rlutcs, rsds rsus, rsdscs, rsuscs, rlds, rlus, rldscs | Radiative fluxes at top of atmosphere and surface. |
| Figure 8 | ua | Zonal wind |
| Figure 9 | ta | Temperature |
| Figure 10 | psl | Sea-level pressure |
| Figure 11 | tas | Near-surface temperature |



**Table A3.** Acronyms, initialisms and abbreviations used in manuscript.

| | |
|---|---|
| AMIP | Atmospheric Model Intercomparison Project |
| CCCma | Canadian Centre for Climate Modelling and Analysis |
| CanAM | Canadian Atmospheric Model |
| CanESM | Canadian Earth System Model |
| CMAM | Canadian Middle Atmosphere Model |
| CERES | Clouds and the Earth's Radiant Energy System |
| CMIP | Coupled Model Intercomparison Project |
| CFMIP | Cloud Feedback Model Intercomparison Project |
| CLASS | Canadian Land Surface Scheme |
| CSLM | Canadian Small Lake Model |
| CRE | Cloud radiative effect |
| CALIPSO | Cloud-Aerosol Lidar and Infrared Pathfinder Satellite Observation |
| DECK | Diagnostic, Evaluation and Characterization of Klima |
| DJF | December/January/February |
| ERA | ECMWF (European Centre for Medium-Range Weather Forecasts) Reanalysis |
| GMMIP | Global Monsoons Model Intercomparison Project |
| GLD | Global Lake Database |
| GOCCP | GCM-Oriented Cloud-Aerosol Lidar and Infrared Pathfinder Satellite Observation (CALIPSO) Cloud Product |
| HITRAN | High-resolution transmission |
| ISCCP | International Satellite Cloud Climatology Project |
| JJA | June/July/August |
| MAM | March/April/May |
| MODIS | Moderate Resolution Imaging Spectro-radiometer |
| SON | September/October/November |
| TOA | Top of atmosphere |

*Author contributions.* JNSC drafted the manuscript and created the figures, contributed to development of CanAM5, and performed simulations with CanAM4 and CanAM5 used in the manauscript. KvS lead the development of CanAM5 and wrote the aerosol section, JL developed the radiative transfer code and wrote the radiative transfer section, JS wrote the tiling, circulation and precipitation sections, DP contributed to CanAM5 development, VA developed CLASS-CTEM and wrote the section that describes it, NM and ML contributed to
5   CanAM5 development, MM developed the Canadian Small Lakes Model and wrote the section describing it, DV developed CLASS and BW processed forcing datasets for CanAM. All authors contributed to writing the manuscript.

*Competing interests.* The authors have no competing interests.



*Acknowledgements.* CERES EBAF data were obtained from the NASA Langley Research Center CERES ordering tool at https://ceres.larc.nasa.gov/data/. GOCCP data were accessed from https://climserv.ipsl.polytechnique.fr/cfmip-obs/Calipso_goccp.html. We thank Carsten Abraham for many helpful comments that improved the manuscript.



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
