# Peer review of "The Canadian Atmospheric Model version 5 (CanAM5.0.3)"

_EGUsphere, 2023_

## Referee Comment (RC1)

*The Canadian Atmospheric Model version 5 (CanAM5.0.3).* J.N.S. Cole and Co-Authors.

The paper documents the scientific structure and basic behaviors of version 5 of the Canadian Atmospheric Model. This model has and will likely continue to advance understanding climate and climate change and is a key participant in international modeling activities in support of ongoing assessment of climate change. The paper is generally clear in its explanations and presents a range of model simulations compared with observations, providing essential information for those using the model or its simulations for a range of scientific and societal impact activities.

*Specific Comments and Questions*

1. Regarding the cloud microphysics factors in Table 1: Are facacc, facaut, and uicefac factors that multiply nominal values for accretion rate, autoconversion efficiency, and ice fall speed? If so, what is the basis for the nominal values? The CanAM5 factors, are quite large, factors of about 10 to 6,000. Especially if these factors scale physically based parameters, is there an issue of physical plausibility? Further explanation and context would be helpful here. Related, on p. 9, l. 35, it is stated that autoconversion rates are not scaled, but the factor scaling efficiency in autoconversion (facaut) in Table 1 is listed as 0.1204. Clarification or additional information is suggested.

2. pp. 12-13, ll. 33-1, Fig. 3: The "notable increase in southern hemisphere low cloud in CanAM5" relative to CanAM4 is not evident on Fig. 3.

3. Fig. 4: What do the solid contours on the two uppermost panels on the right indicate?

4. Figs. 3, 4, 5, 8, 9, 10, and 11: Summary statistics, i.e., mean bias, rmse, correlation coefficients, for the differences between model and observations would be helpful. On Figs. 6 and 7, the bias is evident, but rmse and correlation coefficients would provide valuable additional information about the fidelity of the model patterns to the CERES observations.

5. pp. 18-19, ll. 13-1, Fig. 9: Regarding zonal-mean temperature differences, note that north of 60N in DJF and MAM a large fraction of the space has oppositely signed differences.

6. p. 21, ll. 1-2: The text states that TOA net downward flux was tuned to produce a reasonable 1850 control for CanESM5. Provide a brief characterization of this simulation, i.e., how well is the TOA radiation balanced and what, if any, drifts are occurring?

7. pp. 14-16, ll. 9-4, and p. 21, ll. 1-4: The TOA net LW+SW imbalance ( Earth Energy Imbalance, EEI) in CanAM5 of 3.1 W m$^{-2}$ is quite large relative to the CERES EBAF value of 0.9 W m$^{-2}$ and the IPCC-estimated total anthropogenic 1750-2011 radiative forcing of 2.3 W m$^{-2}$ . This indicates significant errors in the model's ability to simulate the observed energy imbalance of the Earth-atmosphere system given realistic boundary conditions. Retuning to produce a stable coupled integration is effectively a flux adjustment, even if not explicitly applied as such. Alternatively, these tunings can be viewed as introducing compensating errors in the coupled model to correct whatever deficiencies lead to the drift or unrealistic simulations there. The behavior of the coupled model using tunings which produce the observed EEI in an AMIP integration of CanAM5 would provide an informative gauge of the seriousness of these model deficiencies. I would

encourage considering showing a measure of the problematic CanESM5 simulations using an uncoupled  atmospheric configuration with a realistic EEI.   The revised text should acknowledge the importance of these deficiencies.

*Technical Corrections*

p. 2, l. 17: "The the" -> "The"

p. 2, l. 18: "tropopause" -> "troposphere"

p. 4., l. 11: "CanAM5" repeated.

p. 16: l. 13: "shortwave the" -> "shortwave at the"

p. 19, l. 5: Fig. 11 shows CanAM5/4, not CanESM5/2.

---

## Author Comment (AC1)

**Response to Referee #1**

We thank the referee for their review. Below are the original comments in *italics* intermixed with our responses in normal text.

*The paper documents the scientific structure and basic behaviors of version 5 of the Canadian Atmospheric Model. This model has and will likely continue to advance understanding climate and climate change and is a key participant in international modeling activities in support of ongoing assessment of climate change. The paper is generally clear in its explanations and presents a range of model simulations compared with observations, providing essential information for those using the model or its simulations for a range of scientific and societal impact activities.*

*Specific Comments and Questions*
*1. Regarding the cloud microphysics factors in Table 1: Are facacc, facaut, and uicefac factors that multiply nominal values for accretion rate, autoconversion efficiency, and ice fall speed? If so, what is the basis for the nominal values? The CanAM5 factors, are quite large, factors of about 10 to 6,000. Especially if these factors scale physically based parameters, is there an issue of physical plausibility? Further explanation and context would be helpful here. Related, on p. 9, l. 35, it is stated that autoconversion rates are not scaled, but the factor scaling efficiency in autoconversion (facaut) in Table 1 is listed as 0.1204. Clarification or additional information is suggested.*

We agree that the original text was not clear, having mixed factors that scale the process rate (facacc and facaut) and values that are parameters in the formulation (uicefac, which has a value of 770 in the original formulation). We have adjusted the text to clarify these factors and the values.

*2. pp. 12-13, ll. 33-1, Fig. 3: The "notable increase in southern hemisphere low cloud in CanAM5" relative to CanAM4 is not evident on Fig. 3.*

We agree, it is not clear upon rereading the text what feature to which we were referring. This text has been removed.

*3. Fig. 4: What do the solid contours on the two uppermost panels on the right indicate?*

These contours are the zonal mean cloud amount from CanAM5. The caption has been updated to make this clearer,

"Zonal cloud fraction and cloud phase from CanAM5 compared with GOCCP and MODIS observations averaged over 2007-2009. Black contours on the upper and middle plots in the right column are the zonal mean cloud fraction from CanAM5. Bracketed numbers in the bottom row are, respectively, mean bias, root mean square error and Pearson correlation coefficient, computed using data between 75S and 75N.".

*4. Figs. 3, 4, 5, 8, 9, 10, and 11: Summary statistics, i.e., mean bias, rmse, correlation coefficients, for the differences between model and observations would be helpful. On Figs. 6 and 7, the bias is evident, but rmse and correlation coefficients would provide valuable additional information about the fidelity of the model patterns to the CERES observations.*

We thank the reviewer for this suggestion. We have adjusted the figures so that summary statistics are present, which shows for most quantities that CanAM5 performances as well as or better than CanAM4. We only added this information for plots for 2D horizontal field, e.g., top of atmosphere or surface, since the statistics can be difficult to interpret for latitude-pressure plots (Figures 8 and 9).

Information was added as follows,

- Figure 3: Mean bias, rmse and correlation coefficient have been added to the plots. These statistics are computed for data between 60S and 60N to be consistent with the range used for the zonal means.
- Figure 4: Mean bias, rmse and correlation coefficient have been added to the zonal plots. These statistics are computed for data between 75S and 75N to be consistent with the range used for the zonal means.
- Figure 5: Mean bias, rmse and correlation coefficient have been added to the difference plots.
- Figure 6: An additional row has been added to the figure which shows the rmse and correlation coefficients.
- Figure 7: Rather than adding the statistics to these plots, which in our opinion would make it very difficult to read, we have added a new figure based on Figure 7 to the supplemental that shows the mean bias, rmse and correlation coefficient (Fig. R1). The text has been adjusted to refer to this new figure.
- Figure 10: Mean bias, rmse and correlation coefficient have been added to the difference plots.
- Figure 11: Mean bias, rmse and correlation coefficient have been added to the difference plots.
- Figure A1: Mean bias, rmse and correlation coefficient have been added to the difference plots.

[Figure]

*Figure R1. Global mean bias, RMSE and Pearson correlation coefficient for cloud radiative effects.*

*5. pp. 18-19, ll. 13-1, Fig. 9: Regarding zonal-mean temperature differences, note that north of 60N in DJF and MAM a large fraction of the space has oppositely signed differences.*

True.  We have added a sentence to the text noting this,

"However, there are regional and seasonal differences, for example, temperatures between December and May poleward of 60°N in the stratosphere are not as systematically biased warm in CanAM5 relative to CanAM4."

*6. p. 21, ll. 1-2: The text states that TOA net downward flux was tuned to produce a reasonable 1850 control for CanESM5. Provide a brief characterization of this simulation, i.e., how well is the TOA radiation balanced and what, if any, drifts are occurring?*

We have added a reference to Swart et al., 2019 which shows these quantities in the CanESM5 preindustrial control simulation.  As noted in the Swart et al., 2019 paper there are drifts, as are the case for most models participating in CMIP6, but the magnitude of the drifts is significantly smaller than changes over transient CMIP6 experiments.

*7. pp. 14-16, ll. 9-4, and p. 21, ll. 1-4: The TOA net LW+SW imbalance ( Earth Energy Imbalance, EEI) in CanAM5 of 3.1 W m-2 is quite large relative to the CERES EBAF value of 0.9 W m-2 and the IPCC-estimated total anthropogenic 1750-2011 radiative forcing of 2.3 W m-2 . This indicates significant errors in the*

*model's ability to simulate the observed energy imbalance of the Earth-atmosphere system given realistic boundary conditions. Retuning to produce a stable coupled integration is effectively a flux adjustment, even if not explicitly applied as such. Alternatively, these tunings can be viewed as introducing compensating errors in the coupled model to correct whatever deficiencies lead to the drift or unrealistic simulations there. The behavior of the coupled model using tunings which produce the observed EEI in an AMIP integration of CanAM5 would provide an informative gauge of the seriousness of these model deficiencies. I would encourage considering showing a measure of the problematic CanESM5 simulations using an uncoupled atmospheric configuration with a realistic EEI. The revised text should acknowledge the importance of these deficiencies.*

We have adjusted the text to make our point more clearly in Sections 4 and 6.2.  As noted in the original text, when tuning we were targeting a stable climate that satisfies a handful of targets, e.g., the global mean temperature and sea-ice area.  The properties of CanESM5 for a pre-industrial experiment is shown in Figure 1 of Swart et al, 2019.

As part of development, initial versions of the coupled model were tested with a variety of tunings that roughly matched the observed EEI in present-day AMIP simulations of CanAM5.  However, pre-industrial simulations with CanESM5 using such tunings produced colder, unacceptable  global mean near surface temperatures, by up to 2 K, and excessive sea-ice.   While our attempts were not exhaustive, we came to the same conclusion as the reviewer that we were faced with a model structural error, and at the time we were curious what the EEI in present-day AMIP simulations was for other CMIP models.

The reviewer's comment has led us to reconsider the question of just how pervasive this type of structural error is in CMIP6 models. Now that CMIP6 is essentially concluded, we can analyze the output of CMIP6 models to investigate. In the Fig. R2 we have plotted the global mean net downward flux averaged over the period 2003-2009 for all CMIP6 models that have at least one "amip" and one "historical" simulation.  Red dots indicate present-day amip EEI values while black dots indicate historical values.  Interestingly, many CMIP6 models simulate a net flux at TOA that is well outside CERES observations in present-day AMIP configuration (some exceptionally so) but all models are generally less likely to be outside CERES when run in coupled configuration.  From this figure it is clear that such model structural error is quite pervasive across CMIP6 models and that CanESM5 is not an unusual member of this group. As we are not aware of such analysis in the literature, we are motivated to publish these results in a follow-on study to highlight this issue.  We thank the reviewer for reminding us of this question.

[Figure]

*Figure R2. Net flux at TOA averaged over 2003-2009 for CMIP6 models using the "amip" experiment (red) and the coupled "historical" experiment (black).*

**Technical Corrections**

*p. 2, l. 17: "The the" -> "The"*

*p. 2, l. 18: "tropopause" -> "troposphere"*

*p. 4., l. 11: "CanAM5" repeated.*

*p. 16: l. 13: "shortwave the" -> "shortwave at the"*

*p. 19, l. 5: Fig. 11 shows CanAM5/4, not CanESM5/2.*

All of these corrections have been made in the text.

---

## Author Comment (AC2)

**Response to Referee #2**

We thank the referee for their review. Below are the original comments in *italics* intermixed with our responses in normal text.

*This paper is a technical documentation of the atmospheric model CanAM5 that is part of the CanESM used for its CMIP6 simulations. It documented the modifications made to the previous model version CanAM4, which are for the optical properties of cloud particles and land/ocean/snow surface in the radiative calculations, cloud microphysical scheme, and surface processes. It also contains description of the spin-up of the carbon model and tuning of the CanAM5 and CanESM5. The paper then documented the rudimental performances of climatologies in clouds, radiation, zonally averaged winds and temperature, surface pressure and precipitation against observations and CanAM4. It provided some insight, although rather superficial, into the cause of the model biases. I think the paper is valuable for model developers and potential users of its simulations. I therefore recommend a minor revision. The paper is a straightforward documentation. I only have some minor comments.*

1. *For the benefit of the readers, please describe what is the second aerosol indirect aerosol effect that is included in Equation (2). Line 1 on page 5.*

   We have adjusted the text by referencing the second indirect effect and why it is present in CanAM5 but not CanAM4,

   "Along with the new autoconversion parameterization, CanAM5 now accounts for indirect impacts of aerosols on cloud liquid water content and lifetime, i.e., the second aerosol indirect effect (Ghan et al., 2013). This effect was not active in CanAM4 since it used a constant cloud droplet number concentration of 50 cm$^{-3}$ in Eq. 1 (von Salzen et al., 2013).

2. *Line 24 on page 5 missed a period. This is the same in Line 14 on page 7.*

   The periods have been added.

3. *The sentence in line 23 on page 8 is poorly written. Please revise.*

   We have changed the sentence from

   "After finalizing the new and updated physical parameterizations for CanAM5 were finalized, they were held fixed, or frozen, and only a subset of parameters was manually adjusted within a range of physically plausible values to target a stable and reasonable climate in the coupled atmosphere-ocean model CanESM5."

   to

"After finalizing the new and updated physical parameterizations, they were no longer changed, except for a subset of parameters. These parameters were manually adjusted within a range of physically plausible values to obtain an acceptable pre-industrial climate in the coupled atmosphere-ocean configuration of CanESM5 (Swart et al., 2019).".

4. *Line 14 on page 9: add "net downward" after 2.5 W/m2.*

Thank you for pointing this out. We edited the sentence, so it now reads,

"Adjustments required to ensure an acceptable preindustrial climate resulted in a net downward fluxes at TOA (~3.1 W m$^{-2}$) in historical AMIP runs, which is larger than observations."

5. *Line 24 on page 12: "..., there is a shift to more optically thin ( < 23) in CanAM5", this is inconsistent with Figure 2 which shows less optically thin clouds with tau less than 6. It is also inconsistent with Figure 3.*

Reflecting on this comment, we suspect the reviewer might have been interpreting the figures as values from the models, not biases relative to ISCCP observations. Below we reproduce Figures 2 and 3 (Figures R1 and R2, respectively), with the model values instead of the bias relative to observations, which shows our text is consistent with the figures.

We have adjusted text on Figures 2 and 3 and the text referring to the figures to make it clearer that the CanAM evaluation plots are biases to make it more explicit for the readers.

[Figure]

*Figure R1. Reproduction of Figure 2 in text but instead of plotting biases for the three right plots, actual values are plotted.*

[Figure]

*Figure 2. Reproduction of Figure 3 in the text but instead of plotting biases in the right column, the actual values are plotted.*

*6. First paragraph of Section 6.3 on zonal wind: there seems to be cherry picking. CanAM5 shows larger bias than CanAM4 not just in the Northern Hemisphere. It is also larger in the Southern Hemesphere as indicated in Figure 8. The attribution of orographic gravity wave drag parameterization (Line 6-7 in the paragraph) is thus questionable.*

We have attempted to clarify the role of gravity-wave drag and include comment related to SH anomalies as suggested by the reviewer with the change:

"This positive zonal wind bias is associated with changes to adjustable parameter values in the orographic gravity-wave drag parameterization between the two model versions, Sec. 4."

To

"This wintertime positive zonal wind bias is associated with a weakening of the orographic gravity-wave drag due to a change in parameter values between the two model versions (Sec. 4). Similarly, this weakening of the gravity wave drag contributes to a larger positive anomaly of zonal-mean zonal winds in CanAM5 in the Southern Hemisphere wintertime stratosphere."

---

## Referee Report (RR1)

**Review**

*The Canadian Atmospheric Model version 5 (CanAM5.0.3)* J.N.S. Cole and Co-Authors.

p. 4, l. 22: air density units?
p. 5, l. 1: surfaces → surface
p. 8, l. 6: Should it be "If $Z_{snow} \geq Z_{snow,lim}$ "?
p. 11, l. 19: version → versions
p. 12, l. 6: reflectivity it does → reflectivity, doing
p. 12, l. 9: loud → Cloud
Figure 6 legend, last line: mean bias, root → root
p. 19, l. 6: predecessor and → predecessor

---

## Author Response (AR2)

**Response to Referee #3**

We thank the referee for their review. Below are the original comments in *italics* intermixed with our responses in normal text.

*p. 4, l. 22: air density units?*
*p. 5, l. 1: surfaces → surface*
*p. 8, l. 6: Should it be "If $Z_{snow} \geq Z_{snow,lim}$" "?*
*p. 11, l. 19: version → versions*
*p. 12, l. 6: reflectivity it does → reflectivity, doing*
*p. 12, l. 9: loud → Cloud*
*Figure 6 legend, last line: mean bias, root → root*
*p. 19, l. 6: predecessor and → predecessor*

We have implemented all suggested changes, except for the one related to the sign of the inequality for the snow depth. It is correct in the manuscript and is consistent with Verseghy, 2012. If there is fractional snow cover and the snow depth is less than the limit, it is reset to $Z_{snow,lim}$.